# Self-configuring high-speed multi-plane light conversion

José C. A. Rocha [1,2] ✉, Unė G. Būtaitė [1], Joel Carpenter [2] &
David B. Phillips [1] ✉

Multi-plane light converters (MPLCs) – also known as diffractive neural networks – convert an orthogonal set of optical fields into any other orthogonal set via a unitary transformation. MPLC design typically involves optimising a digital model. However, inherently high levels of complexity mean that even a minor mismatch between this model and the physically realised MPLC leads to a severe reduction in performance. Here we create a self-configuring MPLC, converging in minutes while automatically absorbing unknown misalignments and aberrations into the design. To achieve this we introduce 'multi-plane wavefront shaping' – allowing multiple spatial light modes to be reshaped simultaneously. Convergence is accelerated via a high-speed MPLC platform incorporating a kHz-rate phase-only light modulator. Using this approach we demonstrate arbitrary optical transformations and universal mode sorters. Our work paves the way towards ultra-high-fidelity MPLCs with potential applications to optical communications, photonic computing and imaging.

Spatial light modulators (SLMs) are the workhorses of high-dimensional light manipulation[1]. They are capable of arbitrarily patterning a beam of light across millions of independently tuneable pixels[2]. However, despite their high resolution, a single reflection from a planar two-dimensional (2D) SLM can only efficiently transform a single spatial light mode at a time. Yet the next generation of photonic technologies calls for the ability to efficiently modulate an entire basis of spatial light modes simultaneously: deterministically mapping a group of input spatial modes to a new group of output modes. Optical devices that can passively perform such spatial basis transformations have a diverse range of applications. Examples include spatial mode multiplexers for optical communication links[3], multicasting reconfigurable optical switches[4], mode sorters for far-field super-resolution imaging[5,6], light unscramblers for visualising scenes hidden behind opaque media[7,8], and matrix operators in emerging forms of classical and quantum optical computation architectures[9,10].

So why is it not generally possible for spatial basis transformations to be achieved by a single reflection from an SLM? The root of the problem is that a different hologram is typically required to reshape each different mode incident onto an SLM. While these different holograms can be multiplexed and displayed on an SLM together[11],

each mode is diffracted from all multiplexed holograms, resulting in only a fraction of the light being transformed as desired[12]. This limitation affects all 2D planar light manipulation technologies regardless of their resolution, including liquid crystal SLMs, digital micro-mirror devices, deformable mirrors and metasurfaces. To overcome this issue, inherently three-dimensional (3D) light modulation architectures are called for[13]. At present, such technologies are still in their infancy. Photonic integrated circuits (PICs), composed of waveguide arrays with embedded phase shifters on chip, offer a way forward[14–18]. However, PICs are not yet widely available, and difficult to scale up to high dimensions. An emerging alternative technology is free-space multi-plane light conversion, which is the focus of this work.

Multi-plane light converters (MPLCs)[19–23]—which have more recently become known as linear diffractive neural networks[24,25]— consist of a cascade of planar diffractive elements (the 'planes', which here we also refer to as 'phase masks') separated by regions of free-space. Each phase mask imparts a carefully designed spatially-varying phase delay to light flowing through the device, and the diffraction in between each pair of phase masks allows energy to be exchanged laterally. In this way, input optical fields are sequentially processed and transformed into target output fields—emulating a fully 3D light

[1]Physics and Astronomy, University of Exeter, Exeter EX4 4QL, UK. [2]School of Electrical Engineering and Computer Science, The University of Queensland, Brisbane QLD 4072, Australia. ✉e-mail: jd964@exeter.ac.uk; d.phillips@exeter.ac.uk

processing architecture by coarse-graining it into a series of layers. Crucially, MPLCs can efficiently apply distinct transformations to multiple input modes simultaneously, thus achieving the spatial basis transformations that are much sought after in photonics[26].

The design of an MPLC is a non-linear problem—the choice of phase profile on one plane being non-linearly dependent upon the phase profiles on planes further up- or downstream. Therefore, all phase masks must be jointly optimised, which is typically achieved via the process of inverse design[27,28]. A numerical model of the MPLC is iteratively optimised using adjoint methods that, in each iteration, efficiently determine how the phase of all pixels should be adjusted to improve the design[23,29–31]. This process is repeated until the design converges. Once designed, reconfigurable MPLCs can be implemented using multiple reflections from liquid crystal SLMs[25,32–34].

However, as MPLCs are based on cascading planes, they are extremely sensitive to fabrication errors, which accumulate as light propagates through the device. This means that even a minor mismatch between the digital model used in the design phase and the physically realised optical system leads to a severe drop off in real-world MPLC performance[30,35] (as demonstrated in Supplementary Section S8). Implementing an MPLC necessitates pixel-perfect alignment between the phase masks and the propagating fields on every plane, and simultaneous optimisation of tens of alignment degrees of freedom[30]. For optimal performance, a number of factors must be accounted for, including distortion of the input fields, phase aberrations of the planes themselves, and the imperfect response of the SLM (for example, problems arising from surface flatness, lack of parallelism between the optical surfaces within the SLM display, and crosstalk between neighbouring pixels[36]). These issues are exacerbated as the number of planes, and the complexity of their design, increases—holding back the scale of MPLCs and diffractive neural networks that have been successfully demonstrated to date.

To overcome these challenges, it is highly desirable to develop methods to optimise MPLCs and diffractive neural networks in-situ, circumventing the need to precisely match a digital model with the real physical system. Furthermore, granting self-aligning capabilities to free-space MPLCs would be beneficial for their real-world deployment,

enabling high-fidelity operation to be maintained through varying environmental conditions (e.g., temperature changes) that would otherwise risk misaligning these complex optical systems. Advances in this area also push forward the development of physical neural networks that can be trained in-situ[37], and adaptive optical technology capable of reversing the mixing of signals transmitted through complex scattering media—an emerging concept with many future imaging and communications applications[7,8].

In this work we demonstrate a self-configuring free-space MPLC. Despite the large number of parameters to be optimised (up to 32,400 in our experiments), our proof-of-principle device converges on a timescale of minutes using a method in which light is only transmitted in one direction through the optical system. To make this possible, we develop a bespoke optimisation algorithm, and introduce a fast-switching MPLC platform based on a recently developed microelectromechanical system (MEMS)-based SLM[38], shown schematically in Fig. 1a—allowing millions of MPLC configurations to be rapidly explored. This optimisation scheme naturally accounts for the physical characteristics of all system components by absorbing any unknown misalignments and aberrations into the final design. Our work paves the way towards a new generation of high-dimensional and ultra-high-fidelity fast-switching MPLCs.

## Results

### In-situ MPLC optimisation algorithm

We first describe how to automatically configure an MPLC to transform a single input spatial mode to a target output mode (an example is shown in Fig. 1b). Our approach is inspired by methods developed to control the propagation of light through complex scattering media—using a concept known as wavefront shaping[39]. Wavefront shaping can be accomplished by first measuring the transmission matrix (TM) of the medium[40]—a linear matrix operator describing how an arbitrarily shaped field incident on one side of a complex medium will have been reshaped by the time it emerges from the other side. The TM represents a digital model of the medium's optical response, and once known, this model can be used to calculate how the input field should be shaped to generate a target field at the output[41].

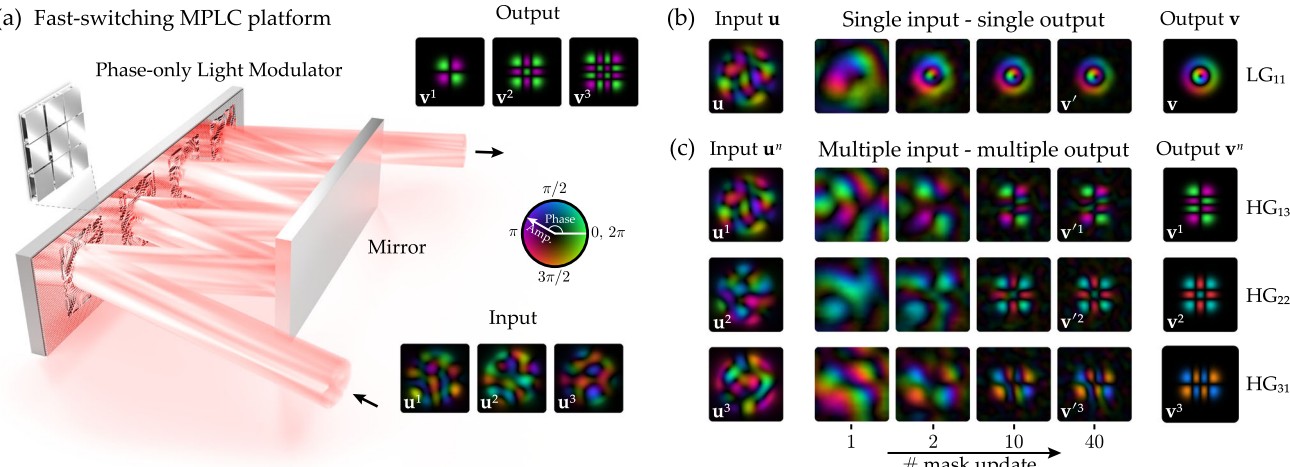

**Fig. 1 | Self-configuring multi-plane light conversion. a** A schematic of a 4-plane MPLC based on a fast switching phase-only light modulator (PLM). Light reflects between different regions of the PLM and an opposing mirror. The PLM micro-mirror heights are optimised to simultaneously transform a set of input modes, such as the three orthogonal speckle modes shown, to a target set of output modes, such as the three Hermite-Gaussian modes at the output. **b** Experimental results showing the automatic in-situ optimisation of an MPLC designed to transform a single arbitrarily shaped input mode (**u**) to a target output mode (**v**), in this case converting a speckle pattern (left most panel) into a Laguerre–Gaussian beam, $LG_{p\ell}$,

with a vortex charge of $\ell = 1$ and radial index $p = 1$ (target mode shown in rightmost panel). The central panels show experimental results of the progression of the output mode throughout the MPLC optimisation process. See Supplementary Movie 1. **c** The same as in **b**, but here showing experimental results of the design of an MPLC to simultaneously transform three input orthogonal speckle modes into three Hermite–Gaussian output modes $HG_{ab}$ of mode order indexed by $a$ and $b$. See Supplementary Movie 2. Supplementary Section S1 shows the fidelity as a function of number of mask updates for the experiments in (**b**) and (**c**).

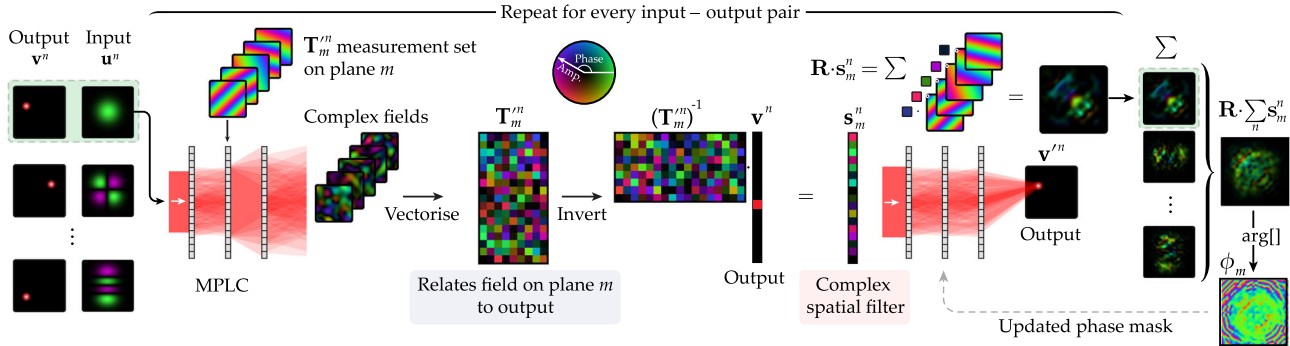

**Fig. 2 | In-situ MPLC optimisation algorithm.** A flowchart depicting the steps to calculate a single phase mask update.

In our case, we treat the MPLC itself as the complex medium. We measure the TM from a particular MPLC plane to the output, and calculate how the phase profile of the plane in question should be updated to generate the desired output field. Once the phase mask is updated, we repeat this process, cycling over each plane in turn until the output field converges. Viewed from the perspective of wavefront shaping, our self-configuring MPLC can be understood as a 'multi-plane wavefront shaper', with the advantage that light can—in principle—be shaped more efficiently[42], and multiple independent modes can be controlled simultaneously, as we show in what follows.

Figure 2 shows a flowchart depicting our in-situ MPLC design protocol. To begin the optimisation, the MPLC is illuminated with input field **u**. In the initial MPLC configuration, field **u** will flow through the optical system generating an output field **v'** that typically has a low correlation with the target output field **v**. Here we represent **u**, **v'** and **v** as column vectors—vectorised versions of the pixelated 2D input and output fields. The MPLC planes are indexed by integer $m$ which takes values from 1 to $M$. We aim to calculate how to update the phase delays imparted by all pixels on plane $m$ to improve the performance of the MPLC.

The propagation of light through the MPLC can be represented by

$$\mathbf{v'} = \mathbf{T}_m \cdot \mathbf{D}_m \cdot \mathbf{H}_m \cdot \mathbf{u}, \tag{1}$$

where matrix $\mathbf{H}_m$ is the TM linking the input field **u** to the field arriving at plane $m$ within the MPLC, and matrix $\mathbf{T}_m$ is the TM linking the field leaving plane $m$ to the output field **v'**. Here $\mathbf{D}_m$ is a diagonal matrix representing how the phase of the light field flowing through the MPLC is modified by plane $m$. We first measure the TM $\mathbf{T}_m$. We sequentially display a set of orthogonal test phase functions on plane $m$—here we display a set of $P$ plane-waves (we also tested Hadamard and 2D discrete cosine functions—see Supplementary Section S7). For each test mode, the corresponding transmitted field arriving at the output (camera) plane is measured holographically. These transmitted fields are vectorised and stacked as columns of $\mathbf{T'}_m$—here the prime indicating that the input basis of $\mathbf{T'}_m$ is different from the pixel input basis of $\mathbf{T}_m$ shown in Eq. (1).

Once measured, $\mathbf{T'}_m$ can be used to calculate the complex spatial filter, $\mathbf{s}_m$, that, if placed at plane $m$ inside the MPLC, would convert the field incident on plane $m$ into the field that will subsequently evolve into **v** at the output:

$$\mathbf{s}_m = \left(\mathbf{T'}_m\right)^{-1} \cdot \mathbf{v}. \tag{2}$$

Here $\mathbf{s}_m$ is a column vector expressing complex coefficients in terms of the plane-wave basis used to measure $\mathbf{T'}_m$. Experimentally we take $\left(\mathbf{T'}_m\right)^{-1} = \left(\mathbf{T'}_m\right)^{\dagger}$, under the assumption that $\mathbf{T'}_m$ is unitary (see Methods). Importantly, $\mathbf{s}_m$ naturally takes into account the unknown shape of the field incident on plane $m$ inside the MPLC ($\mathbf{u'}_m = \mathbf{H}_m \cdot \mathbf{u}$),

which is encoded into the input basis of the measured matrix $\mathbf{T'}_m$. As each MPLC plane can only modify the phase of the light flowing through it (and our aim is to perform a lossless unitary transform using a cascade of phase-only masks), we take the argument of $\mathbf{s}_m$ to obtain the phase mask function $\boldsymbol{\phi}_m$:

$$\boldsymbol{\phi}_m = \arg\left[\mathbf{R} \cdot \mathbf{s}_m\right], \tag{3}$$

where matrix $\mathbf{R}$ transforms the representation of $\mathbf{s}_m$ from the plane-wave basis to the micro-mirror pixel basis (see Methods). Plane $m$ is updated to $\boldsymbol{\phi}_m$, thus improving the MPLC design. This phase mask update constitutes one iteration of our algorithm. We iterate through all $M$ phase masks in this way, and then continue cycling over the planes until the design converges. More than one update of each plane is typically necessary, since when looping back to plane $m$, the phase functions of the surrounding planes have changed, and so further updating plane $m$ can continue to improve the design. Convergence is designated by the change to the phase planes falling below a threshold level, or no further improvement in the fidelity of the output field being observed.

We now expand this design concept to handle $N$ input modes simultaneously—an example of an MPLC transforming $N = 3$ modes is given in Fig. 1c. We label input and output mode pairs with $\mathbf{u}^n$ and $\mathbf{v}^n$ respectively, where $n$ indexes the mode pairs from 1 to $N$. To calculate the updated phase profile of each plane, we illuminate the MPLC with the $N$ input modes in turn, and in each case measure the TM from plane $m$ to the output plane. For example, $\mathbf{T'}_m^n$ is the TM measured from plane $m$ when the MPLC is illuminated with input mode $n$. We calculate a mode pair-dependent set of complex filters $\mathbf{s}_m^n = \left(\mathbf{T'}_m^n\right)^{-1} \cdot \mathbf{v}^n$, and the updated phase function to be displayed on plane $m$ is given by

$$\boldsymbol{\phi}_m = \arg\left[\mathbf{R} \cdot \sum_n \mathbf{s}_m^n\right]. \tag{4}$$

Here the sum over the set of $N$ complex filters $\mathbf{s}_m^n$ serves to find a phase function that multiplexes the action of the phase plane to simultaneously improve the mapping of each input mode to its respective output mode. Supplementary Section S11 provides a high level flowchart detailing all of the steps in our self-configuring MPLC routine.

Our framework can be understood in the context of the wavefront matching method[29]: a coordinate descent based inverse design scheme that is often used to numerically design MPLCs[23]. See, for example, ref. 30 (Supplementary information) for a derivation of the wavefront matching method applied to MPLC design. As in our in-situ MPLC optimisation algorithm, the wavefront matching method also relies on determining the complex spatial filters $\mathbf{s}_m^n$ to calculate how to improve the phase profile on plane $m$. In the wavefront matching method, $\mathbf{s}_m^n$ is found by forward propagating input mode $\mathbf{u}^n$ to plane $m$, backward propagating the target mode $\mathbf{v}^n$ to plane $m$, and comparing

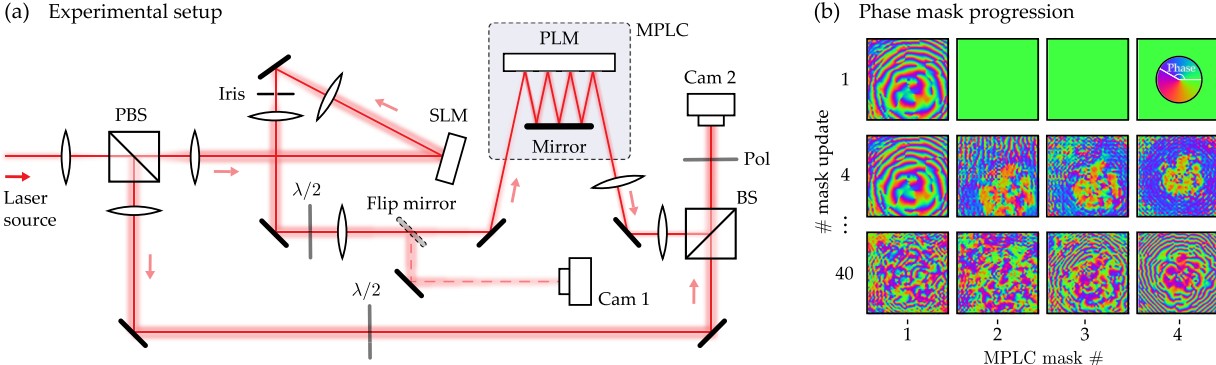

**Fig. 3 | Experimental setup and progression of phase mask design. a** A schematic of our experiment, which is based on a Mach–Zehnder interferometer. A 1 mW linearly polarised laser beam of wavelength $\lambda$ = 633 nm is split into two paths by a polarising beamsplitter (PBS). Light in the upper path is shaped by a liquid crystal SLM (Hamamatsu X13138-01), and transmitted through the MPLC, consisting of a PLM (Texas Instruments DLP6750 EVM) placed opposite a mirror, with a plane spacing of ~6 cm. A flip mirror enables the shaped light incident on plane 1 of the MPLC to be directly imaged (using Cam 1, Basler piA640-210 gm). Light exiting the MPLC is combined with the reference beam (which takes the lower path of the interferometer) via a beamsplitter (BS) and is imaged onto a camera (Cam 2, Basler acA640-300 gm). The field is reconstructed using single-shot off-axis digital holography. **b** Examples of MPLC phase masks displayed throughout the in-situ optimisation procedure—in this case the MPLC is designed to sort 7 orthogonal speckle modes. Top row: first mask update (plane 1). Middle row: MPLC design after four mask updates (planes 1–4). Bottom row: final MPLC design after 40 mask updates (i.e., each of the four planes updated 10 times).

these fields—which represents an efficient adjoint optimisation approach.

It is, in principle, possible to physically achieve both the forward and backward propagation steps necessary for the wavefront matching method to adjointly optimise an MPLC—an approach that is a physical analogue of the error back-propagation algorithm used to train neural networks[43]. For example, ref. 44 has explored this concept through simulations. Indeed, there is much interest in such approaches for in-situ training of physical neural networks[37]. However, our aim here is to avoid the substantial additional complexity and alignment challenges associated with constructing an optical system capable of sending shaped light in both directions (akin to arranging two digital optical phase conjugation systems back to back[45,46]) and accurately holographically imaging the planes inside the MPLC.

In our scheme, light is transmitted only in the forward direction, and we use TM measurement to recover the complex spatial filters $s_m^n$. Reliance on TMs naturally entails making many measurements to calculate each new updated phase function, so our protocol does not classify as an adjoint method. However, since our approach draws inspiration from the wavefront matching method, large changes to the phase mask profiles can be made on each mask update, resulting in optimisation in relatively few mask update cycles. The convergence properties of our algorithm also follow those of the wavefront matching method. In Supplementary Section S2, we show simulations comparing the performance of our self-configured MPLC design method to that achievable via offline design using the wavefront matching method. We find that when the number of optimisation parameters (i.e., $M \times P$) is held constant, the two approaches give the same theoretical performance.

**Fast-switching MPLC platform**

To experimentally implement our in-situ MPLC optimisation routine, we introduce a fast-switching MPLC platform, allowing millions of holographic TM measurements to be made on a practical timescale. We employ a new type of SLM known as a phase-only light modulator (PLM)[38,47–49], shown schematically in Fig. 1a. PLMs are MEMS SLMs consisting of mega-pixel arrays of micro-mirrors. Each micro-mirror can be pistoned vertically with 4-bit precision (i.e., to one of 16 mirror heights), thus controlling the phase of reflected light. Micro-mirror response time is less than 50 µs, resulting in fundamental switching rates of ~20 kHz—although the currently available development models are limited to continuous modulation rates of 1.44 kHz by their

control electronics. The pixel pitch of our PLM model is 10.8 µm, with a pixel fill factor of 94%. Thus it delivers high-efficiency beam shaping on-par with liquid crystal SLMs, and is compatible with the multiple reflections and zero-diffraction order beam shaping of an MPLC architecture. We recently showed how PLMs could be used for high-fidelity wavefront shaping through complex media, and developed bespoke C++ software to synchronise data transfer and continuously display holograms at up to 1.44 kHz[50]. Here we build on this work and program a fast-switching self-configuring PLM-based MPLC.

Figure 3a shows a schematic of our experimental setup, which is based on a Mach–Zehnder interferometer. A collimated laser beam is split into two paths. In the upper path, light first reflects from a liquid crystal SLM which is used to generate the input spatial modes $u^n$ incident on the MPLC. We construct a 4-plane MPLC using a mirror placed opposite the PLM chip (also see Fig. 1a). Four planes were selected in our proof-of-principle experiments as a trade-off between MPLC fidelity, efficiency and optimisation time—see Supplementary Section S6. The transmitted light is imaged onto a high-speed camera which is synchronised with the update cycle of the PLM. The image plane of the camera is located a few centimetres after the final MPLC plane. A reference beam takes the lower path of the interferometer, and is also imaged onto the camera enabling measurement of the fields transmitted through the MPLC via single-shot off-axis digital holography[51].

Since our approach relies on making a large number of interferometric measurements, it is crucial to ensure that the phase drift between the two arms of the interferometer is stabilised within each mask update. Standard phase drift tracking methods (e.g., ref. 52) cannot be directly applied as our scheme relies on the consecutive measurement of TMs with differing MPLC input modes. Therefore, we develop a specific phase stabilisation protocol which is detailed in the Methods. We found this was critical to obtain high-fidelity results.

Prior to commencing an optimisation, it is necessary to define the area of the PLM corresponding to each phase mask. It is enough to roughly estimate the centre of each reflection. No knowledge of the distance between the phase masks, the distance from the last plane to the output camera, or the axial position of the first plane with respect to the incident beams is required. Indeed, our approach is not only limited to free-space MPLCs, but is compatible with any mode-mixing elements placed between the planes. In our experiments, we initialise the phase masks by uniformly setting the phase of all pixels to 0 rad, although any choice of phase mask initialisation can be used—see

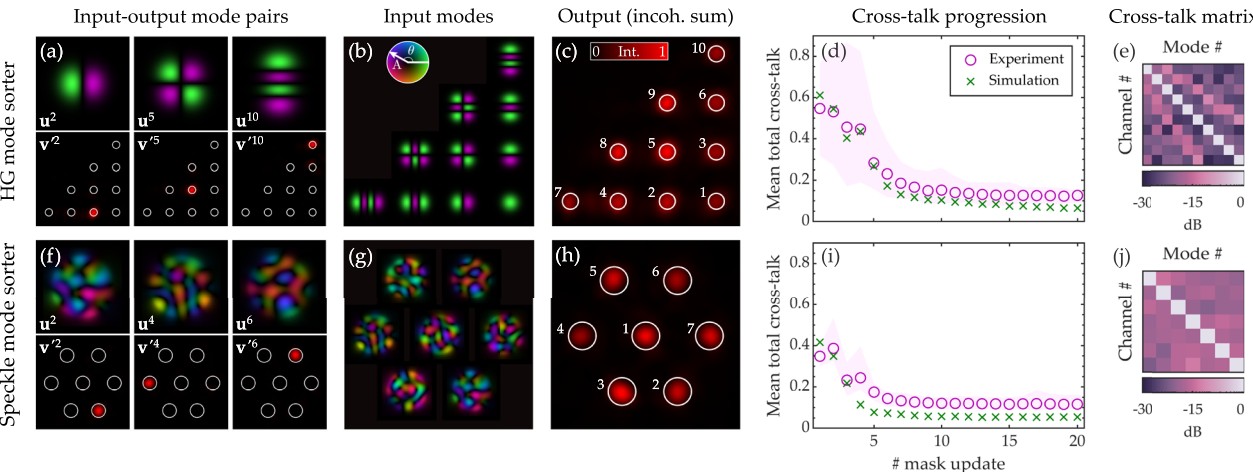

**Fig. 4 | Self-configured Hermite-Gaussian and speckle mode sorters.** Upper panels **a–e** A self-configured 10-mode Hermite–Gaussian (HG) mode sorter. **a** Examples of individual input modes being focussed into specific output channels. **b** All input modes, here shown in the arrangement they will be sorted into. **c** A view of the output channels. Here we plot the incoherent sum of the intensity at the output when the MPLC is illuminated with each mode in turn. **d** The mean total cross-talk throughout the optimisation process ($M = 4$ planes with $C = 5$ cycles yields $M \times C = 20$ mask updates). The mode-dependent cross-talk is given by the total intensity of light transmitted into the wrong output channels, divided by the total intensity of light transmitted into all channels, when the MPLC is illuminated with a given mode. The mean total cross-talk is the mode-dependent cross-talk averaged over all input modes. **e** The cross-talk matrix. Column $n$ shows the intensity of light transmitted into all output channels when the MPLC is illuminated with mode $n$. The average cross-talk is −19 dB. Lower panels **f–j** show equivalent plots to the upper panels, here showing a self-configured 7-mode speckle sorter. In this case the average cross-talk is −17 dB.

Supplementary Section S5 for more discussion on the effect of different initialisations. The number of plane-waves used to sample each TM sets the effective resolution of the corresponding phase mask. Here we tested between $P = 4096$–$8100$ plane-waves, with the range of plane-wave k-vectors chosen to ensure uniform sampling and no aliasing (see Methods).

## Arbitrary field reshaping and universal mode sorting

To test our in-situ MPLC optimisation approach, we first task it with reshaping a single input field to a new target output field. Such reshaping has previously been used, for example, to efficiently generate arbitrary images[53] or couple arbitrarily shaped optical fields into single-mode fibres[54]. Figure 1b shows the mapping of a speckle pattern into a Laguerre–Gaussian ($LG_{11}$) mode. We plot examples of the output field at different stages in the optimisation process, and observe that after 40 mask updates, the fidelity of the output mode reaches 0.95. Supplementary Movie 1 shows the output field as a function of mask update number throughout the optimisation process.

Next, we optimise the MPLC to simultaneously reshape three orthogonal input speckle fields into Hermite–Gaussian modes: $HG_{13}$, $HG_{22}$, and $HG_{31}$, as shown in Fig. 1c. Orthogonal speckles are generated as described in ref. 30. Here slightly lower fidelities of 0.87, 0.92, 0.87 are achieved, respectively, due to the increased complexity of the transformation. The fidelity could potentially be further boosted by increasing the number of test modes used in the measurement of each TM, thus increasing the resolution of the phase masks (see Supplementary Section S2). Supplementary Section S1 shows the fidelity of the output as a function of mask update number for Fig. 1b, c—we see that the MPLC designs have converged after ~20 mask updates. Supplementary Section S10 shows the repeatability of our design process. Supplementary Movie 2 shows the three different output fields generated when the MPLC is illuminated with the three different input fields, as a function of mask update number throughout the optimisation process.

We now turn our attention to spatial mode sorting: redirecting the energy carried by a set of orthogonal input spatial modes to separate locations across a transverse plane at the output. Spatial mode sorters have a variety of future applications in the fields of imaging and optical communications[6,20,55]. In Fig. 4, top row, we demonstrate the optimisation of a 10-mode HG sorter. Figure 4a shows examples of the light from individual spatially overlapping input modes being redirected to separate output channels. Following ref. 23, we arrange the output channels in a triangular lattice, as this configuration has been shown to lead to an efficient HG mode sorter design. All ten input HG modes are depicted in Fig. 4b in the arrangement they will be sorted into. Figure 4c shows the incoherent sum of the output intensities recorded when the HG mode sorter is illuminated with each mode in turn.

Figure 4d shows the mean total cross-talk throughout the optimisation process. Here we compare our experiment to a simulation of an idealised system. We see a similar rate of convergence and generally good agreement between our simulations and experiments. The simulated mean total cross-talk plateaus at a lower value than in our experiments. This is because the simulation represents the best possible case in which the phase function of each mask is continuous (rather than discretised into 16 phase levels as in our experiment), the SLM is 100% efficient, and there is no experimental noise or residual phase drift in the measurements. See Supplementary Section S12 for further discussion on the experimental factors limiting our approach, and possibilities for improvement in future. Figure 4e shows the experimentally measured cross-talk matrix, with an average cross-talk of −19 dB per channel (i.e., the average value of the off-diagonal elements).

Figure 4, bottom row, shows equivalent results for the optimisation of a 7-mode orthogonal speckle sorter[30]—highlighting the universal nature of the spatial transformations that our approach can handle. Here the output spots are arranged into a hexagonal grid. Speckle sorters are examples of arbitrary basis rotations, and have applications in unscrambling light that has propagated through scattering media[7]. In this case, the average cross-talk is −17 dB per channel —higher than the cross-talk for HG mode sorting, since no efficient low plane count MPLC design exists for arbitrary speckle sorting. Supplementary Section S3 shows that reducing the number of sorted speckle fields to $N = 5$ further decreases the cross-talk to −18 dB per channel. Figure 3b shows experimental examples of the phase masks displayed throughout the speckle mode sorter optimisation process.

**Table 1 | Optimisation timescales**

| MPLC type | Figure no. | Inputs (N) | Samples (P) | Opt. params. | Tot. TMs | Opt. configs. | f (Hz) | $d_{TM}$ (s) | $d_{mask}$ (s) | $t_{opt}$ (min) | Proj. $t_{opt}$ (s) |
|---|---|---|---|---|---|---|---|---|---|---|---|
| Speckle to LG | 1(b) | 1 | 4096 | 16,384 | 20 | 89,000 | 720 | 1.5 | 3 | 4 | 9 |
| Speckle to HG | 1(c) | 3 | 4096 | 16,384 | 80 | 354,000 | 720 | 1.5 | 7 | 13 | 27 |
| HG sorter | 4(a–e) | 10 | 4096 | 16,384 | 220 | 970,000 | 720 | 6 | 7 | 47 | 88 |
| Speckle sorter | 4(f–j) | 7 | 8100 | 32,400 | 160 | 1,225,000 | 720 | 10 | 15 | 64 | 122 |
| Speckle sorter | Supp. | 5 | 4096 | 16,384 | 120 | 531,000 | 1440 | 1 | 7 | 10 | 44 |

Optimisation parameters and times for the self-configured MPLCs demonstrated in this work. All MPLCs have $M = 4$ planes, and we show the time to loop over $C = 5$ cycles in each case (i.e., 20 mask updates)—during which all designs converged. Column 5 gives the total number of MPLC parameters to be optimised, given by $M \times P$. Column 6 gives the total number of separate TMs measured during the full optimisation process, given by $(N+1)MC$ for $N > 1$ (see Methods). Column 7 gives the total number of MPLC configurations sampled, rounded to the nearest thousand (i.e., $(1+r_{drift})(N+1)PMC$, for $N > 1$). Column 8 gives the PLM modulation rate used for each design. Columns 9 and 10 give the approximate data processing times in our proof-of-principle implementation. Column 11 gives the optimisation times, in minutes, achieved in our current work. Column 12 indicates the future projected optimisation times, in seconds, for the same parameters if fully-sampling each TM using a next-generation PLM capable of switching at $f = 10$ kHz[48].

## Optimisation timescales

An important aspect of our approach is the time it takes to converge. In our proof-of-principle implementation, the total number of MPLC configurations that need to be tested scales according to $\mathcal{O}(PNMC)$, where $P$ is the number of samples per TM, $N$ is the number of input modes, $M$ is the number of planes, and $C$ is the number of cycles of each plane. More specifically, the time to measure and process the data from a single TM, $t_{TM}$, is given by

$$t_{TM} \sim (1+r_{drift})P/f + d_{TM}, \qquad (5)$$

where $r_{drift}$ is the fraction of extra measurements needed for phase drift tracking (see Methods), $f$ is the SLM modulation rate and $d_{TM}$ is the time required for digital holography data processing (which depends upon the size of the region of interest of the camera and $P$).

The overall MPLC optimisation time, $t_{opt}$, is given by

$$t_{opt} \sim [t_{TM}(N+1) + d_{mask}]MC, \qquad (6)$$

where $d_{mask}$ is the data processing time to create each mask update (which depends upon the size of the mask and $P$). The extra TM measurement is used for inter-TM phase drift tracking (see Methods). A key advantage of our approach is that $\sim P(N+1)$ MPLC configurations (i.e., thousands in this work) can be rapidly sampled without the need for any decision logic to redesign SLM holograms, since calculation of new MPLC patterns only happens at the point of mask update.

The 4-plane, 10-mode HG sorter shown in Fig. 4 used $r_{drift} = 0.08$ and a set of $P = 4096$ plane-waves to measure each TM. In our software implementation, $d_{TM} \sim 6$ s and $d_{mask} \sim 7$ s. Here we operated the PLM at $f = 720$ Hz, which is half of its maximum modulation rate, due to the limited frame-rate of our camera when capturing a larger field of view. This resulted in a TM measurement time of $t_{TM} \sim 12$ s, and so each mask update took $\sim 140$ s. The total optimisation time for $C = 5$ cycles was $t_{opt} \sim 47$ min, which constituted 20 mask updates via the measurement of 220 TMs, achieved by sampling a total of $\sim 970,000$ different MPLC configurations. Table 1 gives the optimisation times (column 11) of all of the self-configured MPLCs demonstrated in this work.

There is scope to substantially decrease these optimisation times in the future. For example, PLMs have a fundamental switching time lower than 50 μs, and models with frame-rates of up to $f = 10$ kHz are currently under development[48]. In addition, the time required for digital holography data processing and phase mask calculation can be markedly reduced using parallelised routines and optimised libraries[56], such that $d_{TM}$ and $d_{mask}$ become negligible. If coupled with higher frame-rate sensors, these improvements would reduce the timescale required to optimise the HG mode sorter we show here from $t_{opt} \sim 47$ min to $t_{opt} \sim 88$ s. Likewise, reshaping of a single input beam

could be achieved in $t_{opt} \sim 9$ s. Column 12 of Table 1 gives projected future optimisation times of all MPLCs demonstrated here if using a next-generation PLM.

In addition to speeding up the PLM frame-rate, we expect it will also be possible to heavily reduce the number of measurements that need to be made. This could be achieved in multiple ways. For example, here we initialise the phase masks with a flat phase function, while if we have some knowledge of the MPLC geometry and are able to use this to coarsely align the system manually with a pre-designed set of phase masks, in-situ optimisation could be used to fine-tune the design. See, for example, refs. 7,57,58 for manual MPLC alignment protocols. Optimising the position of each phase mask has also been accomplished using genetic algorithms[30,32]. Combining our automated approach with these methods could reduce the number of mask update cycles $C$ needed for the design process to converge—as explored in simulations in Supplementary Section S8.

Furthermore, here we have fully-sampled every TM, under the assumption that we have no knowledge about the transfer function of the optical system. However, we know the updated state of each phase mask throughout the optimisation process. Even assuming we have imprecise knowledge of the optical system—such as the geometry and the actual phase delays imparted by the phase masks—this knowledge could be made use of via the framework of compressive sensing[59]. For example, full TMs have been reconstructed using only $\sim 10\%$ of the conventionally required number of measurements by exploiting sparsity priors in the TM structure[60]. Alternatively, adaptive optimisation schemes in place of TM measurement could potentially reduce sampling requirements, at the cost of more regularly updating the phase masks[61,62]. Indeed, our knowledge about the entire optical system steadily increases throughout the optimisation process, as we collect data on the response of the MPLC as a function of micro-mirror state. This information could be used to construct a physically accurate model of the system so that future MPLC designs can be conducted partially or wholly offline. Putting prior knowledge and measured data to good use to speed up optimisation times will be the focus of our future work.

## Discussion

We have introduced a fully self-configuring free-space MPLC rendered feasible by a new type of fast-switching MEMS SLM. Here we have shown MPLC switching rates up to 1.44 kHz, limited by our MEMS PLM control electronics, although PLMs operating at 10 kHz are expected to become available in the near future[48]. Our MPLC platform is not only much faster switching than conventional reconfigurable MPLCs based on liquid crystal SLMs, but is also polarisation agnostic, as shown in Supplementary Section S3. More generally, we note that while reconfigurable MPLCs are highly versatile, they come at the expense of lower

energy efficiency and a larger system footprint than passive fixed MPLCs—which have recently been realised at millimetric scales[63] and have potential to be further miniaturised[64].

We have demonstrated a design protocol inspired by the wavefront matching method[23,29] which optimises the correlation between the target and actual output modes. Our iterative TM-based approach is also compatible with more sophisticated inverse-design schemes[28,31], such as gradient descent-based methods capable of further suppressing modal cross-talk and enabling the trade-off between transform efficiency and fidelity to be tuned[30]—although in this case the number of iterations would increase, extending the optimisation timescale.

In principle, it should be possible to extend in-situ MPLC optimisation to handle multicolour inputs using a tunable laser source enabling input modes of different wavelengths to be transmitted through the MPLC. In this case the resulting wavelength-dependent complex filters $s_m^n$ should be appropriately rescaled when calculating the mask updates. The maximum phase shift that can be applied by a PLM (or liquid crystal SLM) is wavelength dependent, so this should also be accounted for in the design algorithm.

A complication of our approach is that it requires an external reference beam for single-shot holographic output field measurements. To mitigate problems caused by relative optical path length fluctuations, we have developed a new phase-drift stabilisation protocol which tracks and cancels out phase drift (see Methods). Alternatively, our approach is, in principle, compatible with referenceless TM measurement. However, such methods either require multiple output cameras defocused with respect to one another[65], or substantially more measurements (e.g., up to factors of between 7 and 20[66,67]). Furthermore, all of these referenceless techniques require iterative optimisation algorithms to recover output fields, that may be difficult to run at the high modulation rates we rely on in our experiments.

The overall light processing efficiency of an MPLC is given by $\eta = \eta_{\text{design}} \times \eta_{\text{exp}}$. Here $\eta_{\text{design}}$ is the theoretical efficiency of the design, which depends on how many modes the MPLC is tasked with processing, and the nature of the transform—e.g., the 10-mode HG sorter has $\eta_{\text{design}} \sim 40\%$ (see Supplementary Section S2). $\eta_{\text{exp}}$ is the efficiency of the experimentally realised implementation, which depends upon the number of reflections[7]. In our 4-plane prototype MPLC, we estimate $\eta_{\text{exp}}$ ranges between ~3 and 8% (see Supplementary Section S4). Improving the light processing efficiency will be crucial for this technology to transition into real-world applications. Routes to boosting the efficiency include enhancing the reflectivity and optical flatness of the micro-mirrors, and the use of wavelength-optimised anti-reflection coatings on the PLM cover-glass. To a lesser extent, efficiency may also be improved by increasing the pixel fill-factor and increasing the piston bit depth (see Supplementary Section S4 for more discussion).

In this work, our in-situ optimisation algorithm is designed to process a relatively low number of input-output beam pairs (here we show up to ten pairs), and transform these with high fidelity. As our approach mirrors the wavefront matching method, it can, in principle, be scaled up to handle more modes—see Supplementary Section S9. However, optimisation times scale linearly with the numbers of modes and planes in our present algorithm, leading to lengthy optimisation times for large mode counts. Furthermore, some applications require yet larger sets of training pairs, potentially numbering in the thousands[37]. Here alternative in-situ optimisation algorithms may be employed—see ref. 68 for an example applied to optimise a single phase plane. In these cases, new strategies to reduce the measurement overhead of our scheme, as sketched out above, would become essential to render self-configuration practical. Regardless of the algorithm used, the rapid switching rates of our MEMS-based MPLC platform enable exploration of millions configurations in reasonable timescales.

We note that an MPLC preserves the inner product of any two input modes at the output. Therefore, as long as this condition is maintained in the specified MPLC mode transformation, our self-configuring algorithm is compatible with non-orthogonal input and output mode sets. When the inner product between input and output mode sets is not matched (e.g., if attempting to spatially sort non-orthogonal inputs), then it is still possible to optimise an MPLC using our approach, but the design efficiency will be reduced in proportion to the overlap between the inputs[69].

Our protocol makes possible the implementation of MPLCs with unknown and potentially highly complex optical systems between the planes for which there is no physical model available—something that is not possible with the conventional wavefront matching method which relies on an accurate model of the optical system. Being free of a physical model, our concept also opens up possibilities for imaging through highly scattering media—enabling free-space MPLCs that automatically adapt to unscramble strongly scattered light[7]. For example, our scheme does not require knowledge of the shape of input optical fields—only the target output modes need to be specified. Hence our work generalises conventional single-plane wavefront shaping[39,40,70] to multi-plane wavefront shaping. While single-plane wavefront shaping controls the propagation of a single spatial light mode through a scattering medium, multi-plane wavefront shaping grants control over multiple modes simultaneously[71]. Moreover, unlike the multi-conjugate adaptive optics systems developed for astronomy, which are designed to operate under relatively mild levels of volumetric aberration[72], our approach contains no assumptions about the strength of the disorder. Consequently, these techniques may prove useful in emerging multi-conjugate adaptive optics systems designed to ameliorate field-dependent aberrations and enlarge the field of view through strongly scattering media such as biological tissue[73–75].

Finally, we note that self-configuring PICs have been demonstrated recently[8,15,18,76]—including a device with an MPLC-based PIC architecture[77,78]. Our self-configuring free-space MPLC can directly operate on arbitrarily shaped free-space optical fields, and uses an algorithm to optimise a number of parameters that is over two orders of magnitude larger than has been demonstrated using PICs. Nonetheless, the methods we present here are may also have relevance to PIC optimisation, and could facilitate the integration of free-space MPLCs with PICs for ultra-fast operation[79].

In summary, we have demonstrated a path towards the construction of high-dimensional, fast-switching and ultra-high-fidelity free-space MPLCs and linear diffractive neural networks. These versatile optical systems promise exciting future applications across a range of areas, including high-capacity optical communications[3,4], advanced imaging[5,6] and emerging all-optical information processing paradigms[10,26]. Many of these applications call for ultra-high-fidelity multi-dimensional light shaping, and we predict that self-tuning devices will play an important role in achieving this.

## Methods
### Phase drift correction
Since our optimisation approach relies on making a large number of interferometric measurements, it is crucial to ensure that phase drift between the two arms of the interferometer is stabilised. Achieving this is not straightforward, as the optimisation relies on the consecutive measurement of TMs with different input modes. Therefore, we develop a specific phase stabilisation protocol, which is split into two steps: intra-TM and inter-TM phase drift correction.

Intra-TM phase drift correction refers to phase stabilisation within the measurement of a single TM. Here we use a conventional approach of interlacing TM measurements with a standard measurement. The global phase of this standard measurement tracks the phase drift as a function of time throughout the TM measurement. On compiling the TM, the global phase of each TM column is subsequently adjusted to

negate the effect of phase drift. In our experiments, we insert an intra-TM drift measurement after every 12 measurements, which increases the total number of measurements by ~8% (i.e., $r_{drift} = 0.08$). Given the typical modulation rate of $f = 720$ Hz in our experiments, this meant a drift measurement was made at a rate of ~65 Hz, which was much higher than the rate of path length drift between the arms of the interferometer in our case.

Inter-TM phase drift stabilisation corrects the global phase of each of the $N$ TMs measured with different input modes: the $n$th TM from the $m$th plane being labelled $\mathbf{T}''^n_m$. To achieve this, after measuring the first $N$ TMs with different input modes, we create a new input mode which is the sum of all $N$ input modes. We transmit this new input mode through the MPLC system while the $m$th plane displays a plane-wave of index $k$. This results in a scattered field $\mathbf{v}^k_{all}$ arriving at the output camera (Cam 2). This final measurement is related to the earlier TM measurements via

$$\mathbf{v}^k_{all} = \sum_{n=1}^{N} \left( e^{-i\theta_n} \mathbf{v}^k_n \right), \tag{7}$$

where $\mathbf{v}^k_n$ is the $k$th column of TM $\mathbf{T}''^n_m$, and $\theta_n$ is the unknown global phase drift associated with the $n$th TM that we aim to recover−i.e. $\mathbf{v}^k_{all}$ is the sum of the previously measured $\mathbf{v}^k_n$ for all $n$, with each term weighted by the unknown phase drift. Equation (7) can be represented as the matrix equation

$$\mathbf{v}^k_{all} = \mathbf{V}^k \cdot \mathbf{d}^k, \tag{8}$$

where $\mathbf{v}^k_n$ forms the $n$th column of matrix $\mathbf{V}^k$, and $e^{i\theta_n}$ is the $n$th element of column vector $\mathbf{d}^k$. To find the unknown phase drift terms, we rearrange Eqn. (8) to solve for $\mathbf{d}^k$:

$$\mathbf{d}^k = \left( \mathbf{V}^k \right)^{-1} \cdot \mathbf{v}^k_{all}. \tag{9}$$

We note that if the entire transmitted field is not captured, the columns of $\mathbf{V}^k$ are not orthogonal. In this case $\left( \mathbf{V}^k \right)^{-1}$ is given by the Moore–Penrose pseudoinverse of $\mathbf{V}^k$.

In principle, $\mathbf{d}^k$ should be independent of the choice of plane-wave (indexed by $k$) displayed on plane $m$ for the drift calibration measurement. To improve the signal-to-noise ratio of inter-TM drift tracking, in our experiments we take the mean drift phase, averaged over all displayed plane-waves, such that the drift phase associated with the $n$th TM, $\theta_n$, is given by

$$\theta_n = \arg\left[ \sum_k d^k_n \right], \tag{10}$$

where $d^k_n$ is the $n$th element of $\mathbf{d}^k$. Using this approach, a mask update requires the measurement of $N + 1$ TMs. See Supplementary Section S10 for examples of the typical measured levels of intra-TM phase drift in our experiments, and examples of the measurement of $\theta_n$ via our inter-TM phase drift correction process.

## TM sampling
We typically sample the TM with a number of plane-waves that is lower than the number of pixels across each phase mask. Therefore, to ensure each phase profile is uniformly sampled in the plane-wave basis with no aliasing, the maximum transverse component of the plane-wave k-vector is given by

$$k_{max} = \frac{\pi \sqrt{P}}{p \, n_{pix}}, \tag{11}$$

where $p$ is the micro-mirror pitch and $n_{pix}$ is the number of micro-mirrors across one phase mask. For example, in the HG sorter

(Fig. 4a–e), $n_{pix} = 256$ micro-mirrors wide, meaning the total number of pixels per plane is $n^2_{pix} = 65536$. Thus when sampling the TM with $P = 4096$ plane-waves, the final phase masks have an equivalent resolution of $\sqrt{P} \times \sqrt{P} = 64 \times 64$ super-pixels, each of size $n_{pix}/\sqrt{P} = 16$ micro-mirrors (i.e., a patch of $4 \times 4$ micro-mirrors).

To recover the phase mask update function, $\boldsymbol{\phi}_m$, we use Eq. (3) (for $N = 1$) or Eq. (4) (for $N > 1$). Here the matrix $\mathbf{R}$ transforms from the plane-wave basis to the micro-mirror pixel basis. Each column of $\mathbf{R}$ is given by the plane-wave function displayed on plane $m$ of the MPLC during TM measurement: $\exp\left( i(k_x x + k_y y) \right)$, where here $x$ and $y$ denote the lateral Cartesian coordinates of the micro-mirrors, and $k_x$ and $k_y$ specify components of the k-vector of each plane-wave (also noting that $|k| = 2\pi/\lambda$).

## Data availability
The data for the figures in the main text are available in the Open Research Exeter repository under accession code https://doi.org/10.24378/exe.30479279.

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

## Acknowledgements

J.C.A.R. thanks the QUEX Institute for PhD funding (a collaborative enterprise between The University of Queensland and the University of Exeter). J.C. acknowledges financial support from the Australian Research Council (ARC) (FT220100103). DBP acknowledges financial support from the European Research Council (ERC) (ERC Starting grant *PhotUntangle*, no. 804626; and ERC Consolidator grant *ModeMixer*, no. 101170907). D.B.P. and U.G. thank the Engineering and Physical Sciences Research Council (EPSRC) for financial support (EP/Z535928/1). We thank Terry Wright and George Gordon for useful discussions.

## Author contributions

D.B.P. conceived the idea for the project, and developed the in-situ MPLC optimisation algorithm with U.G.B. and J.C.A.R. D.B.P. and J.C. obtained the funding and supervised the project. J.C.A.R. developed all PLM control and self-configuration software, and performed all experiments. U.G.B. and J.C.A.R. performed the data analysis. U.G.B. undertook M.P.L.C. simulations to develop the algorithm and guide the experiments, and performed efficiency measurements with J.C.A.R. D.B.P., J.C.A.R., and U.G.B. wrote the paper with editorial input from J.C.

## Competing interests

The authors declare no competing interests.
