## [Transparent Peer Review file · Nature Communications]

Self-configuring high-speed multi-plane light conversion

Corresponding Author: Mr José Carlos do Amaral Rocha

Version 0:

Reviewer comments:

Reviewer #1

(Remarks to the Author)

The authors report a self-configuring free-space multi-plane light converter (MPLC) that integrates in-situ transmission-matrix (TM) measurement with a fast MEMS-based, phase-only light modulator. By relying solely on forward propagation through the device, their iterative scheme updates each phase plane to map arbitrary input mode sets (speckle, HG, LG) onto target output positions or modes with high fidelity. They present a thorough analysis of optimization timescales, introduce phase-drift correction protocols, and suggest clear avenues for improving efficiency and speed. Before accepted by Nature Communications, the following points should be addressed:

1. Are the fidelity and phase map of the final output pattern consistent when the optimization is repeated multiple times in the same configuration? How strongly does the algorithm depend on the initial phase (e.g., all zeros)?
2. A flowchart illustrating the sequence of multi-mode TM measurements, intra-/inter-TM drift corrections, and phase-plane updates would greatly aid comprehension; It is not obvious whether all input modes (in Fig. 1c) are processed simultaneously or in serial loops, please also indicate it in the flowchart
3. The choice of four planes seems arbitrary. How would performance, speed, or device complexity change if you used fewer or more planes?
4. You employ plane-wave, Hadamard, and 2D discrete cosine functions bases to sample each TM. How does this choice influence convergence speed and final fidelity? Could adaptive or sparsity-driven bases reduce measurement overhead?
5. The current demonstrations handle up to ten modes. How do measurement time, TM inversion cost, and convergence scale for $N > 100$? What are the primary algorithmic or experimental bottlenecks when orthogonality is harder to maintain?
6. How robust is your two-stage drift-correction protocol to real-world temperature fluctuations or vibrations? Must the optimisation restart if drift exceeds threshold, or can it continue seamlessly?
7. All experiments use a single wavelength. What modifications to the optimization workflow or hardware would be required to handle broadband or multi-wavelength inputs?
8. Compared to traditional free-space MPLC implementations, what are the relative advantages or drawbacks in terms of power consumption, system footprint, and component cost?
9. You propose that gain-coated mirrors or pixel-level design optimisations could boost end-to-end efficiency to 20–40%. Please provide supporting simulations or prototype measurements to substantiate these claims.

Reviewer #2

(Remarks to the Author)

In this manuscript, the authors have shown an experimental implementation of a fast, self-configuring, multi-plane light converter (MPLC). Using a MEMS-based SLM for wavefront manipulation, the authors time-efficiently measured the transmission matrices between each plane of the MPLC and the output 'plane' to implement a live version of the wavefront matching algorithm. This method, in principle, should allow them to effectively account for any misalignments, aberrations, and other linear characteristics of the physical system. The techniques used in the work are well documented. This work is original, novel, highly applicable, and extremely relevant to the field, making it potentially suitable for publication in Nature Communications.

However, there is one minor concern that needs to be addressed before I can give my final recommendation. The authors motivated this work by highlighting the difficulties in properly characterising the physical system of the MPLC that leads to a loss of real-world performance. In this regard, does a TM-based MPLC implementation improve the real-world performance achieved by MPLCs relying on physical models? How do the results (cross-talks and fidelities) presented in this work compare to prior art, and what are they (if at all) limited by?

Version 1:

Reviewer comments:

Reviewer #1

(Remarks to the Author)

I can accept the manuscript as it has addressed the reviewers comments.

Reviewer #2

(Remarks to the Author)

The authors have satisfactorily addressed my comments. I strongly recommend that the work be published in Nature Communications.

Reply to reviewers' comments on *Self-configuring high-speed multi-plane light conversion* by José C. A. Rocha et al.

We would like to thank the reviewers for carefully reading our manuscript, and for the time they have dedicated to evaluating our work. We have incorporated their suggestions into the updated manuscript and prepared 8 new supplementary sections, which we believe has strengthened our paper. Below, we respond to the reviewers' comments on a point-by-point basis in purple.

Reviewer 1:

1.0: The authors report a self configuring free space multi plane light converter (MPLC) that integrates in-situ transmission-matrix (TM) measurement with a fast MEMS-based, phase-only light modulator. By relying solely on forward propagation through the device, their iterative scheme updates each phase plane to map arbitrary input mode sets (speckle, HG, LG) onto target output positions or modes with high fidelity. They present a thorough analysis of optimization timescales, introduce phase-drift correction protocols, and suggest clear avenues for improving efficiency and speed. Before accepted by Nature Communications, the following points should be addressed:

Response: We thank the reviewer for their encouraging assessment of our paper.

1.1: Are the fidelity and phase map of the final output pattern consistent when the optimization is repeated multiple times in the same configuration?

Response: We observe that when the optimization is repeated sequentially under similar experimental conditions, it yields similar mask designs, which give a nearly identical level of performance. To demonstrate this, we have added a new Supplementary Section §10, where we show a side-by-side comparison of the final mask designs and the system performance throughout the optimisation process, when experimentally undertaking the same MPLC design twice (3 orthogonal speckles mapped to 3 HG modes). In both cases, we started from the same initial conditions of uniformly flat phase masks. We see that while the final phase masks are not identical, very similar features are shared across the two designs. We put the differences in the design down to the effect of measurement noise and any subtle changes in system alignment leading the optimisation routine down a different design pathway. Despite the small differences in the final mask designs, we find that the system performance (here quantified as output mode fidelity) are very similar throughout the entire design process.

We now refer to this new supplementary section in the "Arbitrary field reshaping and universal mode sorting" section of the main text, where we state:

"SI §10 shows the repeatability of our design process".

1.2: How strongly does the algorithm depend on the initial phase (e.g., all zeros)?

Response: We find that starting the optimisation process from different initial sets of phase masks does lead to differently optimised sets of phase masks which can vary in performance. This behaviour is similar to the conventional wavefront-matching algorithm when started from different

initial conditions. We have added a new supplementary section §5, where we simulate the level of variation in the cross-talk and efficiency of the 10-mode HG mode sorter when starting from flat phase masks and 40 different randomly chosen phase mask sets. In this case we find that MPLC designs initialised with flat phase consistently outperform MPLC designs initialised with random phase masks: while cross-talk is similar when the algorithm is initialised in either way, the design efficiency is improved by $\sim 25\%$ when using a flat phase as an initialisation. We also find that the performance of the randomly initialised phase masks are relatively independent of the particular randomly drawn starting masks.

Experimentally, that there is a technical advantage to starting the optimisation process from flat phase masks rather than random phase patterns. Using flat phase masks means we can direct the majority of the transmitted light onto camera from the start of the optimisation process. In contrast, starting from random phase masks has the tendency to scatter light out of the optical system where it is not measured by the camera, which lowers the signal to noise ratio of the measurements, at least for the first few mask updates. Flat phase masks also represent the most wavelength-independent initial guess, reducing the risk of converging to a highly structured, more wavelength-sensitive solution that is also more susceptible to system drift.

We now refer to this new supplementary section in the “Fast switching MPLC platform” section of the main text, where we state:

“In our experiments, we initialise the phase masks by uniformly setting the phase of all pixels to 0 rad, although any choice of phase mask initialisation can be used – see SI §5 for more discussion on the effect of different initialisations”.

1.3: A flowchart illustrating the sequence of multi-mode TM measurements, intra-/inter-TM drift corrections, and phase-plane updates would greatly aid comprehension; It is not obvious whether all input modes (in Fig. 1c) are processed simultaneously or in serial loops, please also indicate it in the flowchart.

Response: Here we agree with the reviewer that a flowchart would clarify our algorithm. We have now added a flowchart to a new supplementary section §11 (Fig. S12), which details all of the optimisation steps. To answer the specific question: For a particular mask update, the TM associated with each input mode is measured sequentially, and the results are processed together to give a single mask update averaged over all input modes.

We now refer to this new supplementary section in the “In-situ MPLC optimisation algorithm” section of the main text where we state:

“SI §11 provides a higher-level flowchart detailing all of the steps in our self-configuring MPLC routine.”

1.4: The choice of four planes seems arbitrary. How would performance, speed, or device complexity change if you used fewer or more planes?

Response: We chose four planes in this study based on a trade-off between the following criteria: (i) Minimising the optimisation time (which scales linearly with the number of planes); (ii) Achieving relatively low cross-talk (which improves with the number of planes); (iii) Maximising the overall efficiency (which decreases with the number of planes due to reflection losses, despite the fact that the design efficiency itself gradually increases with the number of

planes).

We now discuss this trade-off in a new supplementary section §6, where we also present simulations of the HG mode sorter and speckle mode sorter performance (cross-talk and design efficiency) when varying the number of planes from 2 to 10. We show that although there is improvement in MPLC fidelity beyond 4 planes, it is relatively gradual, thus requiring many more planes to significantly improve the MPLC performance. More generally, the number of planes M needed to efficiently enact an arbitrary transform on N modes typically scales linearly with N [1].

We now comment on this in the “Fast-switching MPLC platform” section of the main paper where we state:

“We construct a 4-plane MPLC using a mirror placed opposite the PLM chip (also see Fig. 1(a)). Four planes were selected in our proof-of-principle experiments as a trade-off between MPLC fidelity, efficiency and optimisation time – see SI §6.”

1.5: You employ plane-wave, Hadamard, and 2D discrete cosine functions bases to sample each TM. How does this choice influence convergence speed and final fidelity?

Response: We found that the plane-wave basis consistently led to the best performance in terms of convergence speed, crosstalk and efficiency. This is because, when measuring a TM with fewer measurements than there are PLM pixels on each plane, the plane-wave basis converges to smoothly varying phase masks – which have also been demonstrated to be optimal for wavefront shaping within forward scattering scenarios such as ours [2, 3]. In contrast, the Hadamard basis will converge to coarsely pixellated phase mask designs which have sharp edges that will scatter light at high angles out of the optical system.

Furthermore, there is a complication when attempting to use the discrete cosine basis: The basis vectors of the plane-wave and Hadamard basis are functions in which only the phase spatially varies, meaning they can be accurately represented using a phase-only spatial light modulator. Conversely, the basis vectors of the discrete cosine basis spatially vary in both their amplitude and phase, and as such cannot be accurately represented using a phase-only spatial light modulator in a lossless manner. This leads to sub-optimal performance of the discrete cosine basis.

We have now added a new supplementary section §7 in which we show in simulation the relative performances of the optimisation algorithm when using different bases to sample the TM, and we further discuss their respective differences. We refer to this supplementary section in the “In-situ MPLC optimisation algorithm” section of the main paper where we state:

“we also tested Hadamard and 2D discrete cosine functions – see SI §7”

1.6: Could adaptive or sparsity-driven bases reduce measurement overhead?

Response: We think that adaptive or sparsity-driven approaches are promising directions for reducing measurement overhead. In the ‘Optimisation timescales’ section of the initial version of the main paper, we raised the possibility of using compressive sensing, which can take advantage of sparsity priors in the TM, where we stated:

“Even assuming we have imprecise knowledge of the optical system – such as the geometry and the actual phase delays imparted by the phase masks – this knowledge could be made use of via, for example, the framework of compressive sensing [59, 60]. This approach has the potential to substantially reduce the number of sequential measurements needed to reliably construct each TM.”

For example, in main paper ref. [60] our group has previously shown that TMs can be reconstructed using as little as 5-10% of the fully-sampled number of measurements by relying on sparsity priors.

As the reviewer mentions, even without using prior knowledge, there is also an opportunity to speed up the design process via adaptive approaches. For example, at each mask update step, our presented method measures the TM which is then used to find the mask m update s_m^n for a particular input mode n . In principle, it should be possible to find a good approximation to s_m^n with fewer measurements by adaptively optimising the phase mask to generate the target n^{th} output mode using methods presented in e.g., refs. [4, 5]. The drawback of these approaches is that the phase masks must be adaptively updated more regularly throughout the optimisation procedure than our present approach. Regularly updating the phase masks is not presently compatible with fast PLM operation (as updated phase masks can only be loaded at a rate of up to 60 Hz).

We have now updated the paper to include these points where we state:

“Furthermore, here we have fully-sampled every TM, under the assumption that we have no knowledge about the transfer function of the optical system. However, we know the updated state of each phase mask throughout the optimisation process. Even assuming we have imprecise knowledge of the optical system – such as the geometry and the actual phase delays imparted by the phase masks – this knowledge could be made use of via, for example, the framework of compressive sensing [59]. For example, full TMs have been reconstructed using only $\sim 10\%$ of the conventionally required number of measurements by exploiting sparsity priors in the TM structure [60]. Alternatively, adaptive optimisation schemes in place of TM measurement could potentially reduce sampling requirements, at the cost of more regularly updating the phase masks [61-62]”.

Integrating such compressive or adaptive strategies would require substantial changes to both the measurement protocol and reconstruction algorithm. Furthermore, our current method does not assume prior knowledge about the mode structure or system sparsity, and as such represents the most general approach. Nonetheless, we agree measurement reduction strategies are worth exploring in future work.

1.7: The current demonstrations handle up to ten modes. How do measurement time, TM inversion cost, and convergence scale for $N > 100$?

Response: We investigated this, and have now included a supplementary section §9 demonstrating, in simulation, the use of our algorithm to design an MPLC that sorts 105 Hermite-Gaussian modes. Akin to the conventional wavefront-matching algorithm, scaling to higher mode counts increases the number of required mask updates for convergence. In this 105-mode design, we found that approximately 80 mask updates were required for reasonable levels of crosstalk and efficiency, which requires over 8,000 TM measurements to reach that point. In the current experimental implementation, the total runtime is roughly evenly split between time in hardware (hologram display – at the PLM rate of 720 Hz or 1.44 kHz) and processing time in software (mainly spent in digital holography). At a hologram rate of 1.44 kHz, we calculate it would take just over 14 hours in hardware for all TM measurements alone. Looking ahead, in the future development of PLM technology, a hologram rate of $f = 10$ kHz could reduce this time to just over 2 hours.

Computational cost of inverting a single TM (i.e., taking its complex conjugate transpose) is the same for both the ten-mode and 105-mode sorter, provided that the camera resolution and the number of basis vectors used to sample the TM are fixed, yet the number of TMs we are required to invert increases linearly with the number of input modes – i.e., for 105 modes, total TM inversion

cost is a factor of ten larger than the case for our ten mode sorter. That said, as we mention in the ‘Optimisation timescales’ section of the main paper, many of the software routines could be optimised benefiting from the highly parallel nature of many of those steps, and with larger mode counts, these optimisations become increasingly worthwhile.

On the same note, the above analysis assumes that we fully sample each TM. As discussed in response to reviewer question 1.6 above, smarter sampling strategies, such as compressive sensing based sparsity-driven approaches, would become increasingly advantageous at this scale. Indeed such strategies are needed to render self-configuration practical for large mode counts.

We now refer to the new supplementary section and comment on the scaling for large numbers of modes in the ‘Discussion and conclusions’ section of the main paper, where we state:

As our approach mirrors the wavefront matching method, it can, in principle, be scaled up to handle more modes – see SI §9. However, optimisation time scales linearly with the numbers of modes and planes in our present algorithm, leading to lengthy optimisation times for large mode counts. Furthermore, some applications require larger sets of training pairs, potentially numbering in the thousands [37]. Here alternative in-situ optimisation algorithms may be employed – see ref. [66] for an example applied to optimise a single phase plane. In these cases, new strategies to reduce the measurement overhead of our scheme, as sketched out above, would become essential to render self-configuration practical.

1.8: What are the primary algorithmic or experimental bottlenecks when orthogonality is harder to maintain?

Response: We now address this point in the ‘Discussion and conclusions’ section of the main paper, where we have added:

“We note that an MPLC preserves the inner product of any two input modes at the output. Therefore, as long as this condition is maintained in the specified MPLC mode transformation, our self-configuring algorithm is compatible with non-orthogonal input and output mode sets. When the inner product between input and output mode sets is not matched (e.g., if attempting to spatially sort non-orthogonal inputs), then it is still possible to optimise an MPLC using our approach, but the design efficiency will be reduced in proportion to the overlap between the inputs [6].”

1.9: How robust is your two-stage drift-correction protocol to real-world temperature fluctuations or vibrations? Must the optimisation restart if drift exceeds threshold, or can it continue seamlessly?

Response: Our phase drift correction protocol has no maximum phase drift limit – it can continue seamlessly throughout any range of global phase drift occurring between the reference arm and the beam transmitted through the MPLC. That said, there is a limit to the *rate* of phase drift we can accurately account for. In our current implementation we measure the relative drift at a rate of 60 Hz. This means we assume that the phase drift occurs on a timescale slower than 1/60 s. In our experiments (on a floated optical table) this condition was easily satisfied. We have now added a new supplementary section §10 demonstrating the typical drift correction under different lab conditions. If phase drift fluctuations may occur more rapidly, the drift measurement rate can be increased up to a maximum of half of the modulator rate (i.e., 720 Hz for our current PLM operating at 1.44 kHz.) at the expense of roughly doubling the total optimisation time (due to all

of the extra drift correct measurements).

For real-world operation, vibrations may also cause changes to laser pointing stability. If laser pointing direction changes appreciably within the measurement of a single TM measurement, then this may introduce substantial errors in the TM. However, when the self-configuring algorithm is running, then it will self-correct (in the following mask updates) for errors due to isolated vibrations causing an erroneous measurement of an individual TM. The main effect of these influences will be to slow convergence time. For example, we found that the optical table can be accidentally bumped and the algorithm will automatically compensate for the resulting measurement errors. The combination of high-frequency global phase drift correction and self-configuration means that the system is highly resilient to external perturbations. To further improve the resilience to perturbations, we envision that instead of measuring the TM on each mask update, adaptive optimisation of each mask update would naturally self-correct for isolated external perturbations [4, 5] – an approach also mentioned in answer to reviewer question 1.6.

We now refer to this new supplementary section §10 in the ‘Phase drift correction’ subsection in the ‘Methods’ section of the main paper, where we have added:

“See SI §10 for examples of the typical measured levels of intra-TM phase drift in our experiments, and examples of the measurement of θ_n via our inter-TM phase drift correction process.”

1.10: All experiments use a single wavelength. What modifications to the optimization workflow or hardware would be required to handle broadband or multi-wavelength inputs?

Response: While our experiments were performed at a single wavelength, the underlying MPLC design is, similarly to the conventional wavefront-matching algorithm, compatible with broadband operation. This could be achieved either by measuring the system response at multiple wavelengths sequentially (at the cost of longer acquisition time) or simultaneously using a broadband source, provided the interferometer arms are well path-length matched. In that case, the existing workflow could be retained, though initial alignment becomes more challenging.

No fundamental changes would be needed to the algorithm itself to handle multi-wavelength inputs – the measurement of each mode at each new wavelength can essentially be treated as a different input mode with appropriate wavelength re-scaling when the mask update is calculated via main paper Eqn. 4.

We have now added a paragraph to the ‘Discussion and conclusion’ section of the main paper where we discuss this prospect:

“In principle, it should be possible to extend in-situ MPLC optimisation to handle multicolour inputs using a tunable laser source that would enable input modes of different wavelengths to be transmitted through the MPLC. In this case the resulting wavelength-dependent complex filters s_m^n should be appropriately rescaled when calculating the mask updates. The maximum phase shift that can be applied by a PLM (or liquid crystal SLM) is wavelength dependent, so this should also be accounted for in the design algorithm.”

1.11: Compared to traditional free-space MPLC implementations, what are the relative advantages or drawbacks in terms of power consumption, system footprint, and component cost?

Response: We first comment on a comparison with alternative reconfigurable MPLC platforms. The system footprint and power consumption of our PLM-based MPLC are similar to conventional

liquid crystal SLM-based MPLC setups. In both cases, the active display requires a power supply with voltages continuously supplied to each pixel of the display. In our implementation, component costs are also comparable. At present, our trial model of TI PLM is considerably cheaper than conventional LCoS SLMs (roughly a factor of ten lower in cost), although this may change when fully packaged versions are rolled out. In the future PLMs may benefit from economies of scale, since they have been developed by TI for next generation mass-market display applications. Our algorithm would – in principle – work similarly with a liquid crystal SLM, though the optimisation would be significantly slower when using standard 60 Hz devices.

Evidently, if the masks never needed to change, the most energy efficient option would be to use fixed phase masks, which once fabricated draw no power. Fixed lithographically etched reflection-based MPLCs can be fabricated with extremely high reflectivity and can be made highly compact. For example some of the authors recently demonstrated a 45-mode, 14-plane reflective MPLC occupying a volume of $2 \times 4.8 \times 1$ mm [7]. Research is ongoing into further shrinking MPLCs using direct laser writing. For example, a 2-mode transmissive device occupying a volume of $50 \times 50 \times 50$ μm has been demonstrated [8]. This is highly promising, however at this scale it is challenging to maintain high fidelity and high mode capacity operation.

We have now added a statement to the ‘Discussion and conclusions’ section of the main paper discussing the above:

“More generally, we note that while reconfigurable MPLCs are highly versatile, they come at the expense of lower energy efficiency and a larger system footprint than passive fixed MPLCs – which have recently been realised at millimetric scales [7] and have potential to be further miniaturised [8]”.

1.12: You propose that gain-coated mirrors or pixel-level design optimisations could boost end-to-end efficiency to 20%-40%. Please provide supporting simulations or prototype measurements to substantiate these claims.

Response: Here we sought to make some comments and projections about the potential efficiency improvements in reconfigurable MPLCs. However since the reviewer recommends this claim should be more strongly evidenced, we have removed this claim from the main paper. We discuss the potential for these improvements in the supplementary §4, where we also now give more detail about how these projections were made.

In summary, our estimates are based on a simple expression using the following equation (also applied in our previous work [9]):

$$\eta_{\text{exp}} = (r_{\text{SLM}}d_{\text{SLM}})^M, \quad (1)$$

where η_{exp} is the experimental efficiency of the physically realised MPLC, which is separated into the product of two contributions:

- Reflection efficiency r_{SLM} – defined as the percentage of incident light reflected from the PLM that is sent to the zero diffraction order when the PLM displays a flat pattern;
- Diffraction efficiency d_{SLM} – the percentage of the zero order light that can be controllably diffracted to the first order, when the PLM displays a phase ramp. The diffraction efficiency can be modelled [10], and decreases for steeper phase ramps, so here we choose an a value of $d_{\text{SLM}} = 0.84$ based on the typical spatial frequencies of the displayed phase patterns in our MPLC designs.

To calculate an estimate for the overall experimental efficiency of the MPLC η_{exp} , the reflection efficiency and diffraction efficiency terms are multiplied and the result taken to the power of the number of phase masks M , which accounts for the compounding of losses due to reflection multiple times.

In our present work, we estimate that $r_{\text{SLM}} \sim 0.63$. For our $M = 4$ plane MPLC, this yields $\eta_{\text{exp}} = (0.63 \times 0.84)^4 \sim 0.08$, which agrees well with our experimental measurements.

Our predictions of possible future increases in efficiency are based on projections of how much the reflection efficiency might be improved in future systems. We assume that diffraction efficiency will remain fixed. We give two examples:

(1) A near infra-red-optimised PLM is currently under development by TI (recently introduced in a presentation at SPIE Photonics West conference in January 2025 [10]), which is designed to operate across the wavelength range $730 < \lambda < 1630$ nm, and is expected to have an improved reflection efficiency of $r_{\text{SLM}} \sim 0.8$. For an $M = 4$ plane MPLC, assuming a reflection efficiency of $r_{\text{SLM}} = 0.8$ suggests an experimental efficiency of $\eta_{\text{exp}} = (0.8 \times 0.84)^4 \sim 0.2$.

(2) Liquid crystal SLMs have undergone many decades of development, and in these cases the reflection efficiency has been improved using narrowband anti-reflection coatings of the cover-glass, and also dielectric backplane mirrors. For example, this has enabled the implementation of liquid crystal SLM-based MPLCs of up to ten planes [11]. These features grant liquid crystal SLMs a reflection efficiency of up to $r_{\text{SLM}} \sim 0.95$. For an $M = 4$ plane MPLC, assuming a reflection efficiency of $r_{\text{SLM}} = 0.95$ suggests an experimental efficiency of $\eta_{\text{exp}} = (0.95 \times 0.84)^4 \sim 0.4$.

We note that the near infra-red PLM mentioned above is not yet available for testing, so we cannot perform any experimental validation of this device. Meanwhile, high efficiency SLMs have been used for MPLCs by other groups, and their reflectivity is well established (e.g., see the specification for the Holoeye PLUTO-2.1-NIR-113 has a quoted reflectivity of 95%, aided by a dielectric mirror backplane and anti-reflection coated cover-glass – see <https://holoeye.com/products/spatial-light-modulators/>).

Reviewer 2:

2.0: In this manuscript, the authors have shown an experimental implementation of a fast, self-configuring, multi-plane light converter (MPLC). Using a MEMS-based SLM for wavefront manipulation, the authors time-efficiently measured the transmission matrices between each plane of the MPLC and the output “plane” to implement a live version of the wavefront matching algorithm. This method, in principle, should allow them to effectively account for any misalignments, aberrations, and other linear characteristics of the physical system. The techniques used in the work are well documented. This work is original, novel, highly applicable, and extremely relevant to the field, making it potentially suitable for publication in Nature Communications.

Response: We thank the reviewer for their favourable comments about our work.

2.1: However, there is one minor concern that needs to be addressed before I can give my final recommendation. The authors motivated this work by highlighting the difficulties in properly characterising the physical system of the MPLC that leads to a loss of real-world performance. In this regard, does a TM-based MPLC implementation improve the real-world performance achieved by MPLCs relying on physical models? How do the results (cross-talks and fidelities) presented in this work compare to prior art?

Response: To answer this question we first consider the difficulties faced in experimentally implementing an MPLC, which fall into two key categories: misalignments and aberrations.

Misalignments: These refer to the relative 3D positions and orientations of the planes with respect to one another and the optical axis of the input light. It is, in principle, possible to manually align an MPLC to correctly match these alignment degrees of freedom with the model used to design the MPLC. Therefore, considering only misalignments, we would expect our algorithm to match a perfectly manually aligned MPLC (as long as the signal-to-noise ratio of the in-situ measurements is high, and enough TM samples were used – see SI §2).

Aberrations: However, if the planes feature unknown aberrations – that are therefore not included in the physical model used to design the MPLC offline – a manually aligned MPLC will always suffer reduced performance. We now include a new supplementary section SI §8 which shows simulations of this effect – we see that even weak aberrations have a significant impact on MPLC performance. In this case our self-configuring MPLC can automatically account for these aberrations, and so will surpass the performance achievable through manual alignment. We now make this point in SI §8.

In our paper we have demonstrated self-configured MPLCs with fidelities of 87-95% and average mode sorting cross-talk ranging from -17 to -19 dB. The fidelity and cross-talk of an MPLC depend upon a number of factors, such as the number of modes being transformed, the number of planes, the distance between the planes, the number of pixels per plane, and the particular transformation enacted. It is therefore not straightforward to make a direct comparison with our current experiments and previously published work – as the above parameters are typically different in each experiment. Nonetheless, we believe the experimentally achieved cross-talks and fidelities that we present are, for our particular set of parameters, on-par with the state-of-the-art examples of well-aligned MPLCs in the literature – see, for example, refs. [12–15].

Furthermore, we emphasize that it is extremely challenging to manually align an MPLC, and

our approach resolves issues that arise if manual alignment is not possible (as seems to have occurred, for example, in ref. [16], where the researchers were limited to 2-plane MPLCs due to problems manually implementing more planes). Indeed, we pursued this line of research in part because of the difficulties we experienced manually implementing a PLM-based MPLC – despite being a lab. with substantial experience in this area (e.g. [9, 12]). We think this may have been due to the unknown surface curvature of our PLM, since MEMS devices typically exhibit lower flatness than liquid crystal SLMs [17]. This meant that designs optimised using the conventional wavefront matching method, via a physical model with no knowledge of the PLM surface curvature, did not well match the physical system. Evidently our self-configuring approach naturally accommodates these unknowns.

Newly added SI §8 illustrates how sensitive MPLCs are to both unknown misalignments and unknown phase aberrations of the planes. In this section we also show the performance of our self-configuring algorithm when initialised from a set of masks that have been pre-designed offline, but are slightly misaligned. Interestingly, in this case we find that our self-configuring algorithm converges more rapidly (roughly twice as fast) than if starting from flat phases, and finds a more efficient final solution than initialising blindly from flat phases. This suggests that first making best efforts to align a pre-designed MPLC, followed by running our self-configuring approach is a promising way to enhance MPLC performance. We now refer to these additional simulations in the ‘Optimisation timescales’ section of the main paper, where we state:

“Combining our automated approach with these methods could reduce the number of mask update cycles C needed for the design process to converge – as explored in simulations in SI §8.”

Furthermore, the ability to automatically optimise MPLCs in-situ, after only basic manual MPLC alignment and irrespective of the nature of the required MPLC mode transformation, offers several other advantages over pre-designed MPLCs. For example, beyond simple aberrations and misalignments, our protocol allows for automatic alignment of MPLCs with unknown and potentially highly complex optical systems between the planes for which there is no physical model available – something that is not possible with the conventional wavefront matching method which relies on an accurate physical model. This represents a capability beyond any other free-space MPLC design approach as far as we are aware. We now make this point in the ‘Discussion and conclusions’ section of the main paper, where we have added:

“Our protocol makes possible the implementation of MPLCs with unknown and potentially highly complex optical systems between the planes for which there is no physical model available – something that is not possible with the conventional wavefront matching method which relies on an accurate model of the optical system.”

Similarly, our algorithm can operate without knowledge of the light modes entering the MPLC, with potential applications to reversing the scrambling of light due to unknown scattering media [9]. We are excited to explore the possibilities opened up by these advances in the future.

2.2: And what are they (*the results*) (if at all) limited by?

Response: Below we consider in more detail what limits the performance of our current experiments, and suggest potential future improvements to mitigate these factors:

(1) In our experiments, the number of TM samples (P) limits the highest spatial frequency of the MPLC masks. Increasing P (up to a maximum of the total number of pixels on each plane) would increase the resolution of the masks and thus improve the MPLC performance (see SI §2).

However this also results in a longer optimisation time. Here strategies to reduce measurement overhead, as discussed in the ‘Optimisation timescales’ section of the main paper and in answer to reviewer point 1.6 above, will be explored in the future.

(2) The constrained field-of-view of our high-speed camera may also place a limit on the MPLC performance. For example, in order to run the camera at a rate that could keep up with the PLM switching rate, we had to reduce the field-of-view, meaning that after digital holographic processing, the resolution of the final fields was relatively low ($\sim 20 \times 20$ pixels). We think that using a camera that can accommodate a larger field-of-view at these high frame rates will improve the measurement fidelity of the output fields, and thus improve the design process.

(3) TM measurement noise will also contribute to inaccuracies in the mask update function. This effect could be reduced by increasing the intensity of the laser used (there is ample scope for this as we used a low power 1 mW HeNe laser in our current experiments), and by using a more sensitive camera with a lower noise floor.

(4) Non-linear effects cannot be captured by a linear TM and so cannot be properly accounted for in our in-situ design process. The main source of non-linear effects is likely to be multiple reflections between the MEMS mirrors and the coverglass in front of them. Such multiple reflections result in a non-linear relationship between PLM mirror heights and the phase retardation of the reflected field. This nonlinearity is structural [18, 19], preserving the wavelength, and should not be confused with optical nonlinearities (e.g. harmonic generation). This is only a problem on the mask from which the TM is being measured – we do not have to be concerned with such multiple reflections on the other masks which are held static and so the effect of multiple reflections can still be captured by a linear matrix operator. We believe this effect is small, but would be further reduced by anti-reflection coating on the PLM coverglass. Furthermore, replacing TM measurement with adaptive optimisation [4, 5], as mentioned in point 1.6 above, would circumvent the need to assume a linear matrix and thus avoid the effect of such non-linearities.

We have now include this information in a new supplementary section §12, which is also referred to in the ‘Arbitrary field reshaping and universal mode sorting’ section of the main text where we state:

‘See SI §12 for further discussion on the experimental factors limiting our approach, and possibilities for improvement in future’

Other minor modifications:

- We corrected Equation 10 by removing the denominator of the summand compared to our original submission. This constant denominator doesn't cause any issues but was unnecessary (as is constant) and had not been implemented in our experiment.
- We changed how we calculate the average cross-talk from the data in the cross-talk matrix. Previously we first converted each off-diagonal value to dB and then averaged these dB values. We now average the off diagonal elements and subsequently convert this average value to dB. This second method slightly increased our quoted average cross-talk, but we think it is a fairer representation of the average cross-talk.

References

1. López Pastor, V., Lundeen, J. & Marquardt, F. Arbitrary optical wave evolution with Fourier transforms and phase masks. *Optics Express* **29**, 38441–38450 (2021).
2. Mastiani, B., Osnabrugge, G. & Vellekoop, I. M. Wavefront shaping for forward scattering. *Optics express* **30**, 37436–37445 (2022).
3. Mastiani, B., Cox, D. W. & Vellekoop, I. M. Practical considerations for high-fidelity wavefront shaping experiments. *Journal of Physics: Photonics* **6**, 033003 (2024).
4. Vellekoop, I. M. & Mosk, A. Phase control algorithms for focusing light through turbid media. *Optics communications* **281**, 3071–3080 (2008).
5. Conkey, D. B., Brown, A. N., Caravaca-Aguirre, A. M. & Piestun, R. Genetic algorithm optimization for focusing through turbid media in noisy environments. *Optics express* **20**, 4840–4849 (2012).
6. Goel, S. *et al.* Simultaneously sorting overlapping quantum states of light. *Physical Review Letters* **130**, 143602 (2023).
7. Fontaine, N. K. *et al.* Wafer scale fabrication of multi-plane light conversion devices in 49th European Conference on Optical Communications (ECOC 2023) **2023** (2023), 1063–1066.
8. Porte, X. *et al.* Direct (3+ 1) D laser writing of graded-index optical elements. *Optica* **8**, 1281–1287 (2021).
9. Kupianskyi, H., Horsley, S. A. & Phillips, D. B. All-optically untangling light propagation through multimode fibers. *Optica* **11**, 101–112 (2024).
10. Orr, G. *et al.* High-volume production test methodology and parametrics of the Texas Instruments Phase Light Modulator (PLM) in Emerging Digital Micromirror Device Based Systems and Applications XVII **13383** (2025), 9–17.
11. Lib, O., Shekel, R. & Bromberg, Y. Building and aligning a 10-plane light converter. *arXiv preprint arXiv:2409.20039* (2024).
12. Kupianskyi, H., Horsley, S. A. & Phillips, D. B. High-dimensional spatial mode sorting and optical circuit design using multi-plane light conversion. *APL Photonics* **8** (2023).

13. Brandt, F., Hiekkamäki, M., Bouchard, F., Huber, M. & Fickler, R. High-dimensional quantum gates using full-field spatial modes of photons. *Optica* **7**, 98–107 (2020).
14. Lib, O., Sulimany, K. & Bromberg, Y. Processing entangled photons in high dimensions with a programmable light converter. *Physical Review Applied* **18**, 014063 (2022).
15. Goel, S. *et al.* Inverse design of high-dimensional quantum optical circuits in a complex medium. *Nature Physics* **20**, 232–239 (2024).
16. Martinez-Becerril, A. C. *et al.* Reconfigurable unitary transformations of optical beam arrays. *Optics Express* **32**, 41111–41126 (2024).
17. Rocha, J. C. *et al.* Fast and light-efficient wavefront shaping with a MEMS phase-only light modulator. *Optics Express* **32**, 43300–43314 (2024).
18. Xia, F. *et al.* Nonlinear optical encoding enabled by recurrent linear scattering. *Nature Photonics*, 1–9 (2024).
19. Yildirim, M., Dinc, N. U., Oguz, I., Psaltis, D. & Moser, C. Nonlinear processing with linear optics. *Nature Photonics*, 1–7 (2024).